# Multimodal Language Learning in the Face of Missing Modalities

## Abstract

We propose *extended multimodal alignment (EMMA)*, a generalized geometric method combined with a cross-entropy loss function that can be used to learn retrieval models that incorporate an arbitrary number of views of a particular piece of data, compounded by the challenge of retrieval when a modality becomes unavailable. Our study is motivated by needs in robotics and computer-human interaction, where an agent has many sensors and thus modalities with which a human may interact, both to communicate a desired goal and for the agent to recognize a desired target object. For such problems, there has been little research on integrating more than two modalities. While there have been widely popular works on self-supervised contrastive learning based on cross-entropy, there is an entirely separate family of approaches based on explicit geometric alignment. However, to the best of our knowledge there has been no work on combining the two approaches for multimodal learning. We propose to combine both families of approaches and argue that the two are complementary. We demonstrate the usability of our model on a grounded language object retrieval scenario, where an intelligent agent has to select an object given an unconstrained language command. We leverage four modalities including vision, depth sensing, text, and speech, and we show that our model converges approximately 5 times faster than previous strong baselines, and out-performs or is strongly competitive with state-of-the-art contrastive learning. The code is publicly available on GitHub and will be included for the camera-ready version (it is redacted for anonymity).

## 1 Introduction

Inspired by the multimodal nature of human interaction with the world, it is intuitive that agents learning about the world, upon encountering new concepts and new objects, should form a model that incorporates information from all available sensors and data sources. The benefits of integrating multiple modalities are twofold; first, complementary information can be extracted from different modalities that can help with understanding the world, and second, additional modalities can help in the cases when one or more sources of data about the world become unavailable. Grounded language understanding, in which natural language is used as a query against objects in a physical environment, allows a real-world, intuitive mechanism by which users can instruct physical agents to engage in tasks such as object retrieval. Visuolinguistic approaches to such object inference tasks typically involve training on large pools of image/text pairs and then using language to subselect elements of the sensed environment (Hong et al., 2021; Zhuge et al., 2021).

Although physical agents such as robots typically have access to sensory and interactive modalities beyond vision, and learning from multiple modalities can improve performance on downstream tasks, most approaches use at most two sensory inputs (e.g., visual data such as RGB plus depth images) with single labels, such as those provided by textual natural language. Simultaneously using additional inputs from different modalities is an underexplored area, in part due to the domain-specific nature of such $n$-ary learning approaches. With the modern proliferation of audio and text based communication and home agents (e.g., Alexa/Google Home), there is a growing need to handle more modalities, and simultaneously their potential failures.

One difficulty with working with complex multimodal data is the increased likelihood that one or more modalities may have missing information. Hardware can become damaged or defective, sensors can get blocked or obstructed, and various adverse but not uncommon conditions can remove a modality from use. Current multimodal approaches are typically not robust to the loss of one or more modalities at test time, as may happen if, for example, a physical agent fails to retrieve data from a particular sensor. In order to fully leverage multimodal training data while being robust to missing information, we propose a generalized distance-based loss function that can be extended to learn retrieval models that incorporate an arbitrary number of modalities.

We consider the domain of grounded language-based object retrieval (Hu et al., 2016; Nguyen et al., 2021), in which objects in an environment must be identified based on linguistic instructions. This can be considered a special case of image retrieval (Huang et al., 2017; Ma et al., 2020; Novak et al., 2015; Vo et al., 2019) in which objects are identified using visual inputs in combination with other sensor modalities. Approaches to acquiring grounded language have explored various combinations of sensor inputs such as depth and RGB with labels provided by textual language or speech (Richards et al., 2020). However, despite the multisensory nature of object retrieval, much of the existing work has not previously been extended to include an arbitrary number of modalities.

The ultimate aim of this work is to take arbitrary input modalities about novel objects, including both spoken and written language, and build a model that allows a system to correctly retrieve that object given a subset of those input modalities. In this work, we propose our new method EMMA, a generalized geometric method combined with a cross-entropy loss function that can be used to learn retrieval models that incorporate an arbitrary number of views of a particular piece of data. We demonstrate steps towards this ultimate goal with a mechanism that learns quickly from a variety of input signals and is robust to ablation of certain inputs. This is a generalization of approaches to grounded language learning in which specific input modalities are labeled with language to allow for future identification, but explicitly seeks to be agnostic about the nature of the individual sensory and linguistic inputs.

Our contributions are as follows:

1. Proposing a new approach to multimodal learning by combining geometric and cross-entropy methods and showing that it outperforms and converges faster compared to each method separately.

2. Our method is designed to be applicable to additional modalities as they become available.

3. Our method converges approximately 5 times faster than previous strong baselines, and out-performs or is strongly competitive with state-of-the-art contrastive learning (Chen et al., 2020) and supervised contrastive learning (Khosla et al., 2020) models when all modalities are available.

4. Demonstrating robustness in the face of missing input signals when one or more modalities are ablated at test time, and proposing a simple averaging method to quantify performance for multimodal object retrieval scenarios.

5. Demonstrating the separate utility of speech and text as sources of label information by treating them as sensory input modalities, rather than explicitly as labels.

## 2 Related Work

There is extensive work on the task of image retrieval, of which physical object retrieval can be considered a special case. In this task, language is used to formulate queries against datasets of images, for example in text-and-image matching tasks for fashion data (Gao et al., 2020; Wen et al., 2021), sketch retrieval (Huang et al., 2017), and general photographs of objects (Ma et al., 2020; Novak et al., 2015; Hong et al., 2021). Prior work has focused solely on language and visual modalities, using text or a combination of language and vision to perform visual retrieval (Vo et al., 2019). While recent works have focused on grounding with respect to a knowledge set (Zheng & Zhou, 2019; Meng et al., 2020), our work focuses on robust multimodal learning to perform grounding over an arbitrary number of input modalities.

The growing number of datasets that contain modalities beyond images and text demonstrates the importance and the challenge of this task. Bagher Zadeh et al. (2018) introduce a large dataset of multimodal opinion sentiment and emotion intensity (CMU-MOSEI) for sentiment analysis and emotion recognition. They also introduce a novel multimodal fusion technique called the Dynamic Fusion Graph (DFG). Kebe et al. (2021) present a multimodal dataset of household objects containing RGB images, depth images, written language, and spoken language, which has been used to support learning grounded language directly from speech given small datasets (Kebe et al., 2022). Baltrušaitis et al. (2019) propose a new taxonomy of multimodal machine learning by introducing five technical challenges in addition to typical early and late fusion categorization. They cover different approaches to multimodal machine learning such as *representation*, *translation*, *alignment*, *fusion*, and *co-learning*. In this work, we focus on alignment, the process of identifying the direct relationship between elements across modalities.

We highlight a key difference with recent work like Jangra et al. (2020) and Hu et al. (2019) who develop multi-modal retrieval models. These retrieval models are class based, where the model is trained to recognize any object of the same retrieval class as being the 'same.' Our method and data is instance based, where different objects are not considered the same even if they have the same class, due to the grounding goals of our work (i.e., the agent should identify the specified object, not equivalent objects). These works also do not consider performance in the case of failure of an input modality.

A number of different approaches have been proposed for linguistic-based alignment learning. Alayrac et al. (2020) use a self-supervised contrastive learning method to learn and combine representations from three modalities of visual, audio, and language. Audio and visual domains are mapped to the same space because they are fine-grained, containing dense information for each frame of video, and these embeddings are then mapped to a coarse space where text is also mapped. Their network respects specificity and comparability of different modalities. The loss function they define consists of two distance-based terms for the corresponding spaces, differing from ours in that they do not handle arbitrarily many modalities and do not focus on robustness in the face of modality drop-outs. Nguyen et al. (2020) take a similar approach and perform pairwise cosine similarity to align images and natural language text descriptions to perform object retrieval tasks. Nguyen et al. (2021) take a cross-modal manifold alignment approach to grounded language learning using triplet loss (Chechik et al., 2010), and perform object retrieval by connecting text descriptions of objects to their corresponding RGB-D images.

In general there are two different families of approaches to multimodal learning in the literature. One family is based on classification and cross-entropy losses such as Khosla et al. (2020); Chen et al. (2020) and the other family is geometric based such as Poklukar et al. (2022); Carvalho et al. (2018); Salvador et al. (2017); Nguyen et al. (2021). In this paper we propose to marry the two families and show that combining geometric approach with cross-entropy method is superior.

The way we formulate our geometric loss function is similar to lifted structured loss (Song et al., 2016), where they take a geometric approach in a unimodal scenario and consider pairwise distances among all items in a batch, but in this paper we apply our idea to four modalities, and our formulation can be extended to any number of modalities. This extended geometric formulation results in a loss function that is similar to Poklukar et al. (2022) where they propose a geometric contrastive learning approach in which they fuse all embeddings as a central embedding and then minimize the distance between each embedding and that central embedding. However, their method does not take into account any notion of classification.

The geometric approach is also known as contrastive learning (Qin et al., 2021), which is based on the idea of similarity and distance of elements of training data (Carvalho et al., 2018; Nguyen et al., 2021; Salvador et al., 2017). Triplet loss is also a special case of contrastive loss (Khosla et al., 2020). Contrastive loss is mostly used in self-supervised learning (Bui et al., 2021; Alayrac et al., 2020; Chen et al., 2020), with some exceptions such as Khosla et al. (2020), who use contrastive loss for a supervised problem. Most of these methods are implemented for a non-multimodal dataset (e.g., RGB images only). Our work is a specialization of contrastive loss that is focused on the multimodal input and language problem.

We note in particular that standard triplet learning approaches often require an expensive *mining* step to find harder negative samples to enable effective learning (Hoffer & Ailon, 2015; Schroff et al., 2015; Ferrari et al., 2018; Zhao et al., 2018; Zhai et al., 2018). This is problematic for throughput and scalability, as

run time becomes quadratic in batch size. Our approach does not require mining to obtain good performance, alleviating these issues along with their complicated history in reproducibility and parameter tuning (Musgrave et al., 2020; Raff, 2021; 2019).

Some of the most closely related work focuses on learning based on more than two input modalities. Veit et al. (2018) train a joint model of images, hashtags, and users to perform image retrieval. Their task is similar to ours—given a description, we want to find the image/object in the scene that best matches the description. They tackle this problem by forming a three-way tensor product model. They use a ranking loss to train the model where the score of an observed triplet is higher than an unobserved triplet. They sample six negative triplets per positive sample triplet, and use each of them as a negative in the loss. The downstream retrieval task is then simply done by taking the argmax of the tensor product for a given user. This work differs from ours in that it becomes computationally complex as more and higher-dimensional modalities are added. Our proposed geometric loss function resembles work on quadruplet loss (Chen et al., 2017; Tursun et al., 2021) but is intended to scale to an arbitrary number of modalities.

To the best of our knowledge, there has been limited work on incorporating more than three modalities into a general learning approach. Lei et al. (2020) use inputs from three modalities of image, sketch, and edgemap to perform image retrieval given the sketch. They use a co-attention mechanism, and their loss function contains alignment loss, cross-entropy loss, and sketch-edgemap contrastive loss. Mittal et al. (2020) use Canonical Correlational Analysis (CCA) to differentiate between ineffective and effective modalities for the emotion recognition task from multiple input modalities including face, speech, and text. Multimodal learning has been used for different applications including but not limited to object retrieval. Abouelenien et al. (2014) use language, physiological response, and thermal sensing to detect deceit. Liu et al. (2017) take two modalities as input and output predictions in a third modality. Tursun et al. (2021) uses quadruplet loss for two modalities only; image and sketch. A quadruplet is composed of a sketch picture as an anchor, a negative example from sketch domain, a negative example from picture domain, and a positive example from picture domain. Chen et al. (2017) also uses quadruplet loss, but does not support multiple domains. In contrast, our method is not restricted to two modalities.

## 3 Problem Description

Given a language command (either text or speech) that describes an object, we want our model to retrieve the correct object from a set of objects. This problem is an exemplar of tasks found in the area of grounded language learning in the fields of robotics and natural language processing. Intuitively, the goal is to take unconstrained natural language queries and select the appropriate object based on the complete set of sensor inputs available to the agent. We demonstrate on a domain containing four modalities, all of which refer to objects in the environment: spoken language, written text, RGB (image) inputs, and depth camera inputs. Figure 1 illustrates a small visualization of our object retrieval task: the spoken query "A white textbook titled algorithms" is provided to our contrastive model, which identifies the item (outlined in red in fig. 1) as the most likely item the query is referring to.

More formally, given a spoken language command $x_s$, a textual language command $x_t$, a set of RGB images $X_r = \{x_r^{(1..n)}\}$, and a set of depth images $X_d = \{x_d^{(1..n)}\}$, the task is to retrieve the correct object by choosing the index that has the minimum distance from either of the language commands across all modalities. Depending on which modalities are or are not ablated, we consider up to four distances: $sr$, a vector of distances between $x_s$ and all RGB images in $X_r$; $sd$, a vector of distances between $x_s$ and all depth images in $X_d$; $tr$, a vector of distances between $x_t$ and all RGB images in $X_r$; and $td$, a vector of distances between $x_t$ and all depth images in $X_d$. In order to select the correct object, we first perform a component-wise average of the relevant modality pair distances for the available modalities. Then, we select the object which had the minimum distance, i.e., we perform an argmin on this average vector of multiple-modality distances. Depending on the missing/available sensors during test time, we might have any combination of these four distances. For example, if no written instructions are available at test time,[1] we compute the component-wise average of $sr$ and $sd$, and then select the object whose coordinate resulted in the lowest average distance.

---

[1]This setting is particularly salient. While large bodies of text are frequently available at training time, a person interacting directly with a physical agent may well prefer to use only spoken instructions.

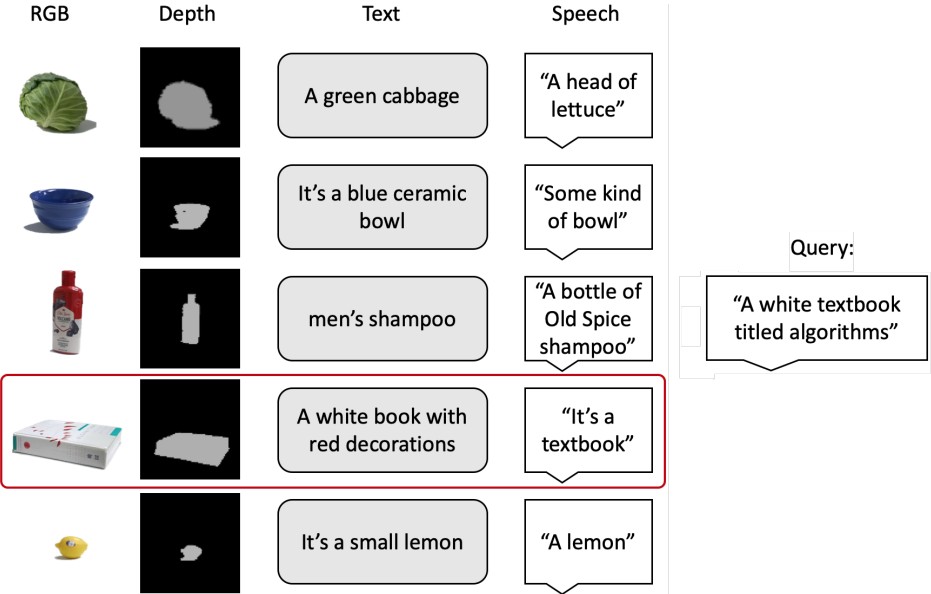

Figure 1: The experimental object retrieval setup, in which objects are represented by four modalities: an RGB image, a depth image, spoken descriptions, and textual descriptions. Given a query in some modality, our approach seeks to select the object that is the best fit, per a trained model. This approach, detailed in section 4, is able to rank objects as to appropriateness even when one or more modalities is ablated at test time (e.g., depth inputs are missing), and outperforms state-of-the-art contrastive learning approaches.

This method allows us to extend our model to support arbitrary modalities while remaining robust to test cases in which some modalities are missing or incomplete.

## 4 Approach

In keeping with previous work on the closely related problem of image retrieval, we focus on contrastive loss approaches, in which the goal is to learn an embedding of data where similar samples—in our case, samples belonging to the same class of object—are embedded 'close' to one another, while dissimilar samples are farther apart. We develop a novel geometric loss function, GEOMETRIC ALIGNMENT, that simultaneously minimizes intra-class distances and maximizes inter-class distances across each pair of modalities, yielding a model that is both effective at the retrieval task defined above and robust to modality dropouts at test time. We take an additional step and combine this GEOMETRIC ALIGNMENT loss with a classification-based (cross-entropy) loss function which results in a superior model compared to either of geometric or cross-entropy losses alone; we refer to this combination as Extended Multi-Modal Alignment, or EMMA.

### 4.1 Baselines

We compare both EMMA and GEOMETRIC ALIGNMENT against contrastive learning (Chen et al., 2020) and supervised contrastive learning (Khosla et al., 2020), which for conciseness we refer to as SUPCON. We consider SUPCON as the main baseline since it is a general version of multiple contrastive loss functions including triplet loss, the traditional version of contrastive loss usually used in self-supervised settings (Chen et al., 2020), and N-pair loss (Sohn, 2016).

#### 4.1.1 Contrastive Loss

We compare our model against the contrastive learning method presented by Chen et al. (2020) where they use the normalized temperature-scaled cross entropy loss (NT-Xent). In order to implement this loss function, we use cosine similarity as suggested in the SimCLR paper (Chen et al., 2020). Another possibility

is to use an inner dot product (Khosla et al., 2020); if not normalized, this can lead to numerical instabilities and overflow/underflow since the dot product is not bounded, but the result is the same whether we use normalized inner dot product or cosine similarity. Contrastive loss function is formulated in eq. (1):

$$-\sum_{i \in I} \log \frac{\exp(sim(z_i, z_{j(i)})/\tau)}{\sum_{a \in A(i)} \exp(sim(z_i, z_a)/\tau)} \tag{1}$$

where $i$ is the index of anchor, $j(i)$ is the index of positive item with respect to the anchor $z_i$ and is not the same as anchor, $A(i)$ is the set of all negatives and the one positive indices excluding anchor, and $z = f(x)$.

We can treat different modalities of the same instance as additional input for that instance that augment the available information, and consider them as the positive points for the anchor. Equation (1) can be rewritten with the sum over more than one positive item as formulated in eq. (2):

$$-\sum_{i \in I} \sum_{p \in P(i)} \log \frac{\exp(sim(z_i, z_p)/\tau)}{\sum_{a \in A(i)} \exp(sim(z_i, z_a)/\tau)} \tag{2}$$

where $I$ is a batch consisting of one or more instances each with a set of all its modalities, and $P(i)$ is the set of modalities/augmentations of the anchor $i$ excluding itself (e.g. RGB image, depth image, speech, text) and $z = f(x)$. Therefore, if we have 4 modalities and the batch size is 64, the size of $I$ is 256, the size of $P(i)$ is $M - 1 = 3$ where $M$ is the number of modalities, and the size of $A(i)$ is $256 - 1 = 255$.

### 4.1.2 Supervised Contrastive Learning

Khosla et al. (2020) extend the contrastive learning method (NT-Xent) and propose a supervised way of performing contrastive learning to treat not only augmentations of the anchor, but also every item that shares the same label with anchor as positives. This loss function is shown in eq. (3).

$$\sum_{i \in I} \frac{-1}{|P(i)|} \sum_{p \in P(i)} \log \frac{\exp(z_i \cdot z_p/\tau)}{\sum_{a \in A(i)} \exp(z_i \cdot z_a/\tau)} \tag{3}$$

Although this loss function does not use cosine similarity, embeddings are normalized before performing dot product, which is equivalent to cosine similarity.

The main difference between the contrastive loss baseline in section 4.1.1 and SupCon is that there is no notion of meaningful negative points in contrastive loss, and everything in the batch that is not the anchor or one of the positive views is considered to be negative. In SupCon, however, all elements in the batch that have same label as the anchor are also considered positives, in addition to different views of the same instance. While the denominators of eqs. (2) and (3) stay the same, this subtle difference affects the numerator and includes more positive examples, which prevents the unintended use of actual positives as negative examples.

While this model is a strong baseline, the authors applied it to a unimodal dataset. In this paper we extend this baseline to work with a multimodal dataset and show that it is slower than EMMA to learn and its performance is also lower when all modalities are available during test time.

Since SupCon considers all pairwise distances in each batch, with $M$ modalities and a batch of size $B$ each batch contains $B \times M$ items, and SupCon computation involves $(BM)^2$ pairwise distance terms which is dependent on batch size. However, the computations of our Geometric Alignment approach is agnostic with respect to batch size which makes it scalable.

### 4.2 EMMA: Extended Multimodal Alignment

Our proposed multimodal method is composed of two complementary parts. The first part is a geometric loss based on distances in the latent space, and the second part is a contrastive loss based on cross-entropy. The geometric loss is faster to learn while the cross-entropy method is more aligned with the downstream task of object retrieval. Hence we propose to combine them.

**Geometric Alignment Loss**  We define a distance-based loss function which can be used for an arbitrary number of modalities. Our proposed method is inspired by the well-known similarity-based triplet loss (Carvalho et al., 2018; Nguyen et al., 2021), and is similar to contrastive loss (Chen et al., 2020; Khosla et al., 2020) under some settings. Triplet loss-based learning works by forcing similar concepts from different domains 'together' in some shared embedding space, while forcing dissimilar concepts 'apart.' It is so named because it relies on three data points from the training set: a positive, a negative, and an anchor point. However, standard triplet loss cannot be used for more than two modalities.

To address this issue, we use pairwise distance optimization for all data points. Our method can be used for an arbitrary number of modalities. In contrast to our work, previous works that use triplet loss (Kebe et al., 2021; Nguyen et al., 2021) concatenate RGB and depth to form a single "vision" vector to handle three modalities, so they cannot robustly handle RGB or depth sensor ablation during test. Moreover, our method has the advantage that it does not require providing hard negative examples. Therefore, we modify the concept of triplet loss as follows. During training, we sample two different instances and their corresponding representations from all modalities into two sets—one positive set (referring to a specific object) and one negative set (referring to some different object) as shown in fig. 2. Unlike some prior triplet loss methods (Kebe et al., 2021; Nguyen et al., 2021), the anchor is not randomly chosen from different modalities in each batch. Instead, in our setting, every item in the positive set becomes anchor once, and we minimize the distance between that item and other items in the positive set, while minimizing the distance between that item and all items in the negative set. It can be seen as an one-to-many relationship instead of an one-to-two relationship in the triplet loss formulation. We consider the following terminology:

- Positive (Instance): A set of embeddings of one data point (e.g., an RGB image of an apple, corresponding depth image, text description, and speech description of the same apple) as shown with green rectangles in fig. 2

- Negative (Instance): A set of embeddings of another data point of a different object (e.g., an RGB image of a mug, corresponding depth image, text description, and speech description of the same mug) as shown with orange rectangles in fig. 2

- Anchor (Modality): Every modality of the positive set is chosen as the learning anchor once. In our formulation, anchor is a "concept" referring to the points in the positive sets rather than a single instance in itself. In fig. 2, each of the 4 modalities is selected as an anchor once. The anchor is used as the basis for learning distances between positive and negative samples.

The objective is then to first minimize the distance between each pair of positive points from heterogeneous modalities, and second, maximize the distance between each positive and negative points from all modalities. We refer to this approach as GEOMETRIC ALIGNMENT which is formulated in eq. (4). An illustration of this loss function is provided in fig. 2.

$$L = \sum_{m_1=1}^{M} \left[ \sum_{m_2=1}^{M} \left[ -\max(dist(z_{m_1}^+, z_{m_2}^-) + \alpha, 0) \right] + \sum_{m_3=m_1+1}^{M} \left[ \max(dist(z_{m_1}^+, z_{m_3}^+), 0) \right] \right] \quad (4)$$

In eq. (4), $M$ is the number of modalities, the superscripts $+$ and $-$ represent positive and negative objects, $\alpha$ represents enforced margin between each positive and negative points which we set to 0.4 for all modalities without tuning, and $z$ is the embedding we get by applying a mapping function $f$, which in our case is a neural network on our input data. In other words, $z_m = f_m(x_m)$, where each modality $m$ has a specific model $f_m$ that is different from the models for other modalities. These models do not share their weights.

Cosine similarity is the opposite of distance, and we need to reverse the logic for maximization and minimization. There are different options to measure distance in embedded space. We use cosine similarity between pairs of embeddings, i.e. we measure the cosine of the angle between embeddings. Cosine similarity is a good choice for high-dimensional data as it is bounded between -1 and 1. Other distance metrics, such

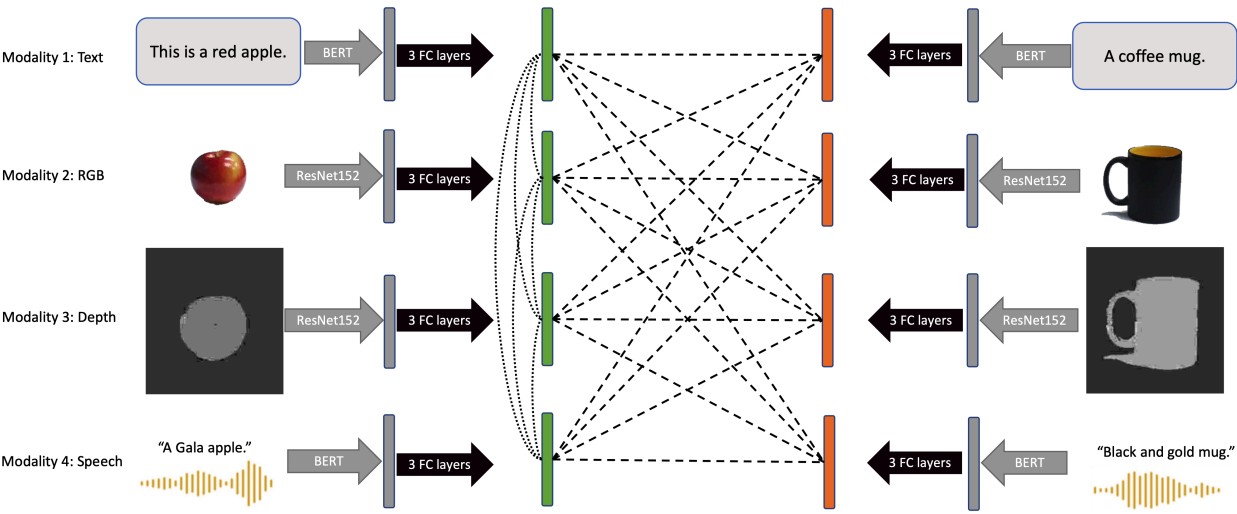

Figure 2: A high-level prototype of our approach and the distances used in the SMALLCAPS GEOMETRIC ALIGNMENT loss with four modalities. Gray arrows indicate pre-trained models that are frozen (i.e. the parameters are fixed and are not trained). The black arrows show 3 fully connected layers with a ReLU activation function (Nair & Hinton, 2010) after the first two layers. These networks are trained. Orange rectangles are negative embeddings and green rectangles are positive embeddings. Dashed lines indicate distances to be maximized while dotted lines indicate distances to be minimized.

as Euclidean distance, grow in value with respect to their dimensionality, resulting in a very large distances for data points.

Here, the generic *dist* function is replaced with the specific $\cos(\cdot)$, and we omit the max notation for clarity by defining eq. (5):

$$
\begin{aligned}
g(x, y) &= \max(\cos(x, y) - 1 + \alpha, 0) \\
h(x, y) &= \max(1 - \cos(x, y), 0).
\end{aligned}
\tag{5}
$$

The first portion of the following equation maximizes all unique pairwise distances between modalities of positive and negative instances. The second portion minimizes the unique pairwise distances among the modalities of positive instances.

$$
\mathcal{L} = \sum_{m_1=1}^{M} \sum_{m_2=1}^{M} g(z_{m_1}^+, z_{m_2}^-) + \sum_{m_1=1}^{M} \sum_{m_3=m_1+1}^{M} h(z_{m_1}^+, z_{m_3}^+)
\tag{6}
$$

Our proposed SMALLCAPS GEOMETRIC ALIGNMENT loss function in eq. (6) can be rewritten as shown in eq. (7) by fully specifying the summations to better understand how our objective function can be reduced to well-known losses such as triplet loss and pairwise loss.

$$
\mathcal{L} = \sum_{i=1}^{M-1} \sum_{j=i+1}^{M} h(z_i^+, z_j^+) + g(z_i^+, z_j^-) + g(z_i^-, z_j^+) + \sum_{i=1}^{M} g(z_i^+, z_i^-).
\tag{7}
$$

If $M = 2$ which means the number of modalities is 2, and we ignore the last two terms in the derived objective function, it results in the triplet loss method. If $M = 2$, then our objective function reduces to the

quadruplet loss method (Tursun et al., 2021; Chen et al., 2017) if we multiply the first term by 2, ignore the third term, and change the last summation to be up to $M - 1$ (which results a single term). If $M = 1$, only the last term remains in the loss function which is exactly the pairwise distance-based loss function. This loss function can be seen as a contrastive loss usually used in the domain of self-supervised learning (Chen et al., 2020). However, our proposed loss function has two advantages over the traditional contrastive loss expressed in eq. (1). The first advantage is that our loss function does not loop over multiple positives and negatives in a large batch. Instead we sample only two objects (positive and negative) each of which have $M$ modalities which gives us $2M$ datapoints (or embeddings). Hence, our model can be trained using smaller batch sizes and reduces the number of negative samples we need. The second advantage is that this loss function can be used in a multimodal setting with an arbitrary number of modalities, and is not limited to a single data type (e.g. RGB images) which is the most common usage of contrastive loss. Although our GEOMETRIC ALIGNMENT is technically quadratic in terms of number of modalities, we observe that experimentally, training time increases only by 10 minutes with each additional modality.

Altogether, our proposed GEOMETRIC ALIGNMENT function contains $3M^2 - M/2$ terms: $M(M-1)/2$ anchor-to-positive distance minimizations and $M^2$ anchor-to-negative distance maximizations. It is noteworthy that our training procedure does not perform any stochastic dropout of modalities to obtain test-time robustness to missing modalities. Moreover, our approach does not need to compute the distance between all items in the batch, as opposed to SupCon.

**Combining Geometric and Cross-Entropy Losses**   The main difference between GEOMETRIC ALIGN-MENT and SupCon is that in GEOMETRIC ALIGNMENT we focus on the geometric interpretation of similarity using cosine distances, while in SupCon, cosine distances are used to compute a classification-based loss function similar to cross-entropy. Both methods are valid and have some advantages that the other method does not offer. In GEOMETRIC ALIGNMENT, the advantages are an intuitive learning objective in terms of distance, interpretability of the learned embedding space, and faster convergence. The advantage of SupCon is that it uses a classification objective which is aligned with the downstream task.

We propose to combine the GEOMETRIC ALIGNMENT defined in eq. (6) with SupCon defined in eq. (3), and we refer to this approach as EMMA, for extended multimodal alignment. The combination is done by taking the sum of the equations for GEOMETRIC ALIGNMENT and SupCon which is shown in eq. (8). When

$$
\mathcal{L} = \sum_{i \in I} \Bigg[ \Big[ \sum_{j=1}^{M-1} \sum_{k=j+1}^{M} h(z_{i,j}^+, z_{i,k}^+) + g(z_{i,j}^+, z_{i,k}^-) + g(z_{i,j}^-, z_{i,k}^+) + \sum_{j=1}^{M} g(z_{i,j}^+, z_{i,j}^-) \Big] +
$$
$$
\Big[ \sum_{m=1}^{M} \frac{-1}{|P(i,m)|} \sum_{\beta \in P(i,m)} \log \frac{\exp(z_{i,m} \cdot z_\beta / \tau)}{\sum_{\gamma \in A(i,m)} \exp(z_{i,m} \cdot z_\gamma / \tau)} \Big] \Bigg],
\tag{8}
$$

where $A(i,m)$ includes all items in the batch except for the $z_{i,m}$ itself and $P(i,m)$ includes all the modalities of all instances that have the same label as current instance excluding $z_{i,m}$ itself. In other words, the collection of embeddings indexed by $P(i,m)$ is $\{ \bigcup_{r \neq m \in M} z_{i,r}, \bigcup_{l \neq i \in I, y_i = y_l} \bigcup_{m=1}^{M} z_{l,m} \}$.

Compared to pure SupCon, combining these loss functions results in faster convergence and slightly improved performance when all modalities are available, and maintains improved performance when modalities are missing. Experimental results are presented in detail in section 6.

## 4.3   Network Architecture

Transformers have become *de facto* architectures in the natural language processing community and have shown great success across different tasks. Similar to Kebe et al. (2021), we use BERT (Devlin et al., 2019) embeddings contained in the FLAIR library (Akbik et al., 2019a;b) to featurize textual input, and wav2vec2 (Baevski et al., 2020) to extract audio embeddings from speech. Both of these encoders output a 3072-dimensional embedding vector which is generated by concatenating the last four hidden layers of their

corresponding networks. FLAIR has historically been used for different natural language processing tasks such as named entity recognition (NER) and part-of-speech tagging (PoS), and wav2veq2 has supported a a number of audio processing tasks, most notably automated speech recognition. Both BERT (Devlin et al., 2019) and wav2vec2 (Baevski et al., 2020) are self-supervised language models using transformers (Vaswani et al., 2017). To process images, we use ResNet152 (He et al., 2016) for both RGB and depth images which gives us a 2048-dimensional embedding vector. Depth images are colorized before passing to the ResNet152.

We then use different instantiations of the same multi-layer perceptron (MLP) consisting of 3 fully connected layers with ReLU activation (Nair & Hinton, 2010) to map each of these embeddings to a shared 1024-dimensional space where we can compute the distances between all embeddings. We note that these MLP networks are distinct and do not share any weights.

## 5 Experiments

In this section we evaluate the quality of object retrieval models learned using the EMMA loss function. We first describe the dataset we use, then describe the metrics by which we evaluate performance, the setup of the experiments, and the baselines against which we compare. We end by presenting and analyzing results.

### 5.1 Data

We demonstrate the effectiveness of our approach on a recent publicly available multimodal dataset called GoLD (Kebe et al., 2021), which contains RGB images, depth images, written text descriptions, speech descriptions, and transcribed speech descriptions for 207 object instances across 47 object classes (see fig. 2). There are a total of 16,500 spoken and 16,500 textual descriptions. The original GoLD paper uses raw RGB and depth images in which other objects are present in the background. We use a masked version of the images where the background is deleted (this masked version converges faster, however, masked and unmasked versions of the GoLD data converge to the same performance). Speech is converted to 16 Hz to match the wav2vec2 speech model.

### 5.2 Setup

To evaluate our model we measure different performance metrics on a retrieval task where the model has to select an object from a set of objects given a language description. Only one of the objects corresponds to the description and the rest are from different object classes. Similar to Khosla et al. (2020), we use a stochastic gradient descent (SGD) optimizer with momentum (Ruder, 2016) with a flexible learning rate starting at 0.05. All models are trained for 200 epochs with a batch size of 64 on a Quadro RTX 8000 GPU. We used a temperature of 0.1 for training the contrastive learning method described in section 4.1.1, and a temperature of 0.07 for training SUPCON as described in section 4.1.2.

To evaluate the performance, we compute the distance between the given natural language description and 5 randomly selected objects (1 of which corresponds to the description, with the others from different object classes). We compute the distance between the language embedding and all available sensory modalities of all candidates as described in section 5.4. In case we have RGB and depth, we compute the distance between language embedding and all candidate RGB embeddings, and we compute the distance between the same language embedding and all candidate depth embeddings corresponding to the RGB embeddings. We then take average of these two distance matrices. Instead of choosing an empirical threshold beyond which objects are considered to be 'referred to,' we choose the closest image embedding (average distance of RGB and/or depth from language) as the prediction. In order to use cosine *distance*, we have to subtract the cosine of the *angle* between two embeddings (which represents similarity) from 1: that is, we compute $1 - \cos(e_1, e_2)$.

### 5.3 Metrics

The best metric to capture the performance in such a scenario is mean reciprocal rank (MRR, eq. (9) for $Q$ queries). For each query we predict the rank of all objects based on their distance from the language command, and then the inverse rank of the desired objects in all queries are averaged. For example, if the

model predicts the desired object as the first rank, then MRR $= \frac{1}{1} = 1$ which means a perfect score, and if it predicts the correct object as the fourth rank among five objects, then MRR $= \frac{1}{4} = 0.25$.

$$\text{MRR} = \frac{1}{|Q|} \sum_{i=1}^{|Q|} \frac{1}{\text{rank}_i} \tag{9}$$

While MRR is more meaningful when it comes to ranking in retrieval tasks, in the real-world scenarios where a robot is asked to hand over an object, if it fails, it does not matter whether the correct object was ranked second or last and the whole system would be considered a failure. Accuracy and micro F1 score are the same in this task, since for each prediction we either have a true positive and no false positives and no false negatives, or we have no true positives, one false positive and one false negative. MRR is a more informative metric because it captures the idea that having the correct object as the second choice should be considered better than having it as a last choice, while in accuracy the score is "all or nothing"—either 0 or 1. Because our approach is designed to be robust to missing information across modalities, we also report MRR and accuracy for different combinations of modality dropouts.

## 5.4 Modality Ablation

We consider an experiment in which we incorporate RGB, depth, speech, and written language to train the model. The loss function requires no changes beyond adjusting the value of $M$ in eq. (4) according to the number of modalities available during training. Our goal is the non-trivial downstream prediction task: determining what objects are being referred to by arbitrary language from a small number of examples. When we consider only text, RGB, and depth, written language is used as the query modality, and we compute the distance of RGB and depth modalities from it and then average them. However, when speech is incorporated as an additional fourth sensory modality, we have three possible choices. First, we could compute the distance of RGB and depth from text and from speech which gives us 4 distance matrices, and then take average of these four. Second, we could treat speech in a similar way to RGB and depth: compute the distance of RGB, depth, and speech from text, and then take an average of three of them. Third, we could compute distances similarly to the first method, but add the distance between language and speech as well and then take the average of 5 distance matrices.

Of these, the first method is the most appropriate choice for a robust multimodal alignment approach. The second and third options are possible during training, but in real-world object retrieval scenarios, having only one form of language instructions is a reasonable scenario—people are not likely to *both* speak about *and* type in instructions for an agent. At test time, depending on which modalities are available to the model, we can use speech, text, or both to compute the distance of RGB and depth embeddings from the linguistic query, and then take the average.

There are total of nine possible cases of modality dropout and the corresponding distance computations. In all these cases $t$ represents text, $s$ represents speech, $r$ represents RGB, $d$ represents depth, and $K$ is the final distance—a matrix if there are multiple language queries and a vector if there is one query. If we only have two modalities, we simply compute the distance between those two modalities; this corresponds to four of the cases; $K_{tr}$ when speech and depth are missing, $K_{sr}$ when text and depth are missing, $K_{td}$ when speech and RGB are missing, and $K_{sd}$ when text and RGB are missing. If we have three modalities, we need to take average of two distances. There are four cases with three modalities, $K_{trd} = \frac{K_{tr} + K_{td}}{2}$ when speech is missing, $K_{srd} = \frac{K_{sr} + K_{sd}}{2}$ when text is missing, and so on. When we have all modalities available we take average of four distances, $K_{tsrd} = \frac{K_{tr} + K_{td} + K_{sr} + K_{sd}}{4}$.

Figure 3 shows the relative performance of EMMA and GEOMETRIC ALIGNMENT against state-of-the-art methods when different modalities are ablated.

| Methods | speech/depth | speech/RGB | text/depth | text/RGB | text/speech/depth | text/speech/RGB | speech/RGB/depth | text/RGB/depth | all |
|---|---|---|---|---|---|---|---|---|---|
| Geometric | 76.82±0.34 | 78.34±0.29 | 89.64±0.38 | 91.13±0.73 | 89.21±0.45 | 90.95±0.83 | 79.37±0.29 | 92.29±0.51 | 92.14±0.45 |
| SupCon | **78.18**±0.58 | **79.69**±0.54 | 89.04±0.88 | 90.56±0.74 | 88.75±0.66 | 90.5±0.69 | **81.2**±0.39 | 91.96±0.42 | 92.03±0.7 |
| EMMA | 77.63±0.29 | 78.66±0.64 | **89.87**±0.5 | **91.26**±0.86 | **89.66**±0.36 | **90.97**±0.66 | 80.32±0.45 | **92.71**±0.5 | **92.72**±0.47 |
| Contrastive | 71.74±0.73 | 73.37±0.39 | 89.72±0.54 | 90.82±0.37 | 89.13±0.61 | 90.26±0.58 | 74.96±0.44 | 91.92±0.41 | 91.72±0.53 |

(a) Average and standard deviation of mean reciprocal rank (MRR) on a held-out test set. MRR is from $\frac{1}{\text{number of objects}}$% to 100%. In most cases EMMA has a better MRR compared to other methods, and all methods have a higher MRR when text modality is present. This shows that text modality is a rich source of information compared to speech, and that EMMA is strongly competitive with state of the art approaches even when relying on speech for the language signal.

| Methods | speech/depth | speech/RGB | text/depth | text/RGB | text/speech/depth | text/speech/RGB | speech/RGB/depth | text/RGB/depth | all |
|---|---|---|---|---|---|---|---|---|---|
| Geometric | 61.95±0.55 | 64.34±0.53 | 82.03±0.57 | 84.6±1.1 | 81.08±0.81 | 84.0±1.4 | 65.84±0.63 | 86.41±0.83 | 85.94±0.74 |
| SupCon | **64.17**±0.92 | **66.52**±1.07 | 81.05±1.22 | 83.65±1.4 | 80.58±1.12 | 83.54±1.23 | **68.7**±0.66 | 86.06±1.21 | 85.82±1.29 |
| EMMA | 63.54±0.53 | 65.07±1.01 | 82.78±0.97 | **85.07**±1.42 | **82.16**±0.64 | **84.37**±1.23 | 67.69±0.81 | **87.38**±0.71 | **87.15**±0.72 |
| Contrastive | 54.82±1.4 | 57.27±0.64 | **82.88**±0.88 | 84.35±1.01 | 81.55±0.93 | 83.26±1.02 | 59.38±0.6 | 86.31±0.67 | 85.75±0.87 |

(b) Average and standard deviation of accuracy (Acc) on a held-out test set. Accuracy is from 0% to 100%. Although accuracy is a strictly less forgiving metric than MRR, these results demonstrate that EMMA still outperforms existing approaches in the majority of cases. When we drop the text modality, the accuracy of the contrastive method is slightly better than random, while EMMA is about 10% more accurate than the contrastive method.

Table 1: Average and standard deviation of mean reciprocal rank (MRR) and accuracy (Acc) over 5 runs with 5 different random seeds on a held-out test set with different modalities ablated during testing. Table 1a shows MRR scores and table 1b shows accuracy. Higher is better for both metrics. For 5 objects, a random guess would have MRR of 0.33 and accuracy of 0.5, and the worst case performance would have MRR of 0.2 and accuracy of 0.0. The batch size is 64 and optimizer is SGD for all experiments. Column headers refer to modalities that are present at query time. Bold numbers represent the best-performing method. EMMA is either better than or very close to both state-of-the-art methods for most of the cases. We observe the same pattern in both MRR and accuracy except the fact that accuracy scores are lower than MRR.

# 6 Results

In this section we provide quantitative and qualitative results by comparing our method against supervised contrastive learning (Khosla et al., 2020) and naïve contrastive loss (Chen et al., 2020). Table 1 summarizes the performance of all models with different ablation metrics. To provide a better sense of the performance measure, consider a model that always ranks the correct object in the second place: Such a model would have an MRR of $1/2 = 0.5$.

Figure 3a Shows that EMMA learns faster and results in a better performance compared to both SupCon and contrastive learning (Chen et al., 2020) when trained using all modalities and with all modalities available during test. We observe that not only does contrastive loss learn more slowly, but that it is prone to overfitting; while this can be addressed with careful tuning of the learning process, an approach that is innately robust to overfitting without tuning is preferable.

When we drop the text modality (fig. 3b), we can see that the performance decreases from about 0.93 to about 0.82, showing that speech cannot completely replace text. In fig. 4 (see section 6), the alignment of shared embeddings for a randomly sampled set of classes is visualized for all four modalities under consideration, suggesting that the speech modality is not aligned as well as the text modality. For this reason, when we drop text and use speech as the main query, the performance decreases. This supports our hypothesis that a geometric alignment of the latent space is crucial to a good performance for object retrieval and multimodal understanding.

In fig. 3b, we observe that when speech is used as the query and the text modality is ablated, the SupCon baseline works slightly better than EMMA, although EMMA still learns faster. The reason is that SupCon optimizes for the classification task, and since the speech modality is less well aligned, using Geometric

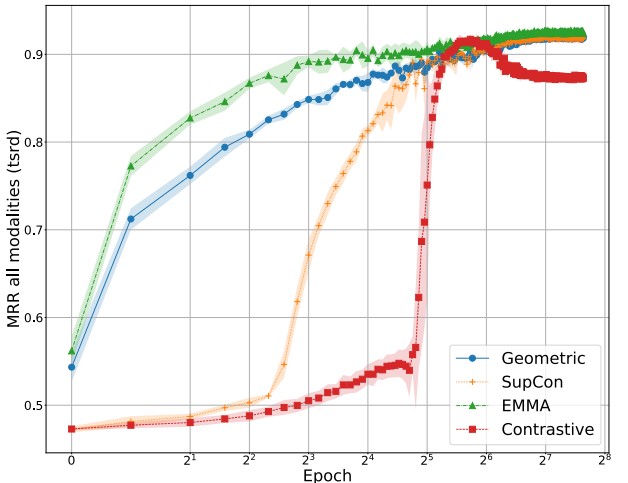

(a) Mean Reciprocal Rank (MRR) on the held-out test set when all modalities are available.

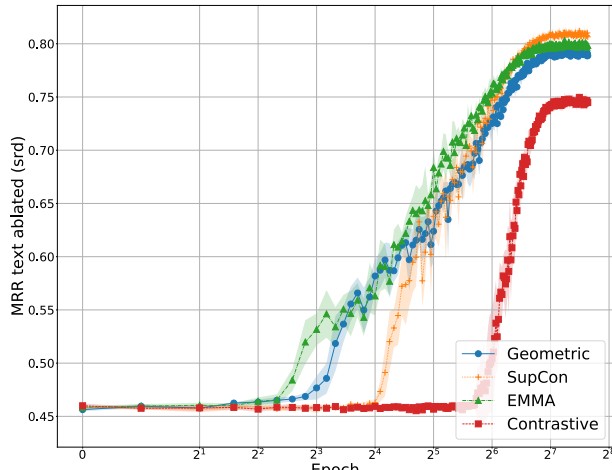

(b) Mean Reciprocal Rank (MRR) on the held-out test set when the text modality is ablated.

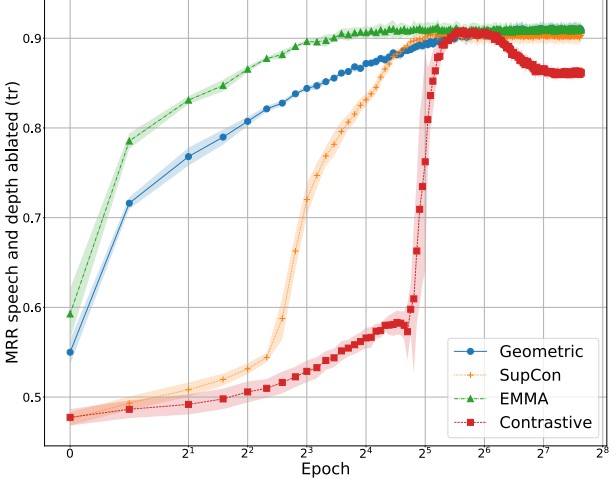

(c) Mean Reciprocal Rank (MRR) on the held-out test set when speech and depth modalities are ablated.

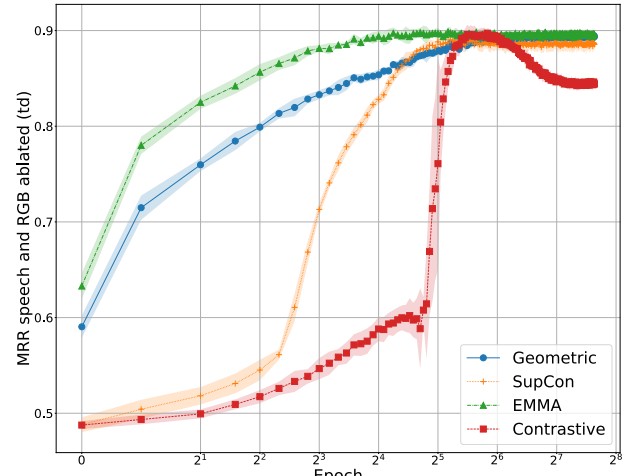

(d) Mean Reciprocal Rank (MRR) on the held-out test set when speech and RGB modalities are dropped.

Figure 3: Mean Reciprocal Rank (MRR, eq. (9)) on the held-out test set with the ablation of selected modalities, all averaged over 5 runs for the downstream task of object retrieval. Red is self-supervised contrastive learning which is prone to overfitting, orange is supervised contrastive learning, blue represents our proposed GEOMETRIC ALIGNMENT, and green is our proposed EMMA loss function. Higher is better. (a) shows the MRR when all modalities are available, (b) shows the MRR when text is removed at test time, (c) shows MRR when speech and depth are ablated, and (d) shows MRR when speech and RGB are removed. We train all models for 200 epochs.

ALIGNMENT makes the downstream task more difficult by trying to pull and push similar and dissimilar data points, respectively. Future research will consider strategies to align more chaotic modalities.

There is very little gap in performance when depth or RGB are dropped in figs. 3c and 3d compared to when we have all modalities in fig. 3a, showing that our model is robust when RGB or depth sensors fail. Also, when depth is dropped in fig. 3c, performance decreases less compared to when RGB is dropped in fig. 3d. This suggests that depth is less informative when compared to RGB, which is consistent with existing vision research results.

**Qualitative Results**   In order to help visualize the performance of learned embeddings, we consider projections of a randomly selected subset of classes of the high-dimensional learned embeddings into a 3-dimensional space using t-SNE (Van der Maaten & Hinton, 2008), a dimensionality reduction technique to visualize high-dimensional data. T-SNE creates a probability distribution over pairs of high-dimensional data where similar pairs have a higher probability and dissimilar pairs have a lower probability. A similar probability distribution is also defined over pairs of data in the lower dimension (either 2D or 3D), and T-SNE minimizes the KL divergence between these two probability distributions. Figure 4 shows the projection onto 3D space to give a better view of the location of embeddings. Although these projections are not perfect, combined with the quantitative results, they demonstrate that our model is learning to map instances of the same class closer to each other regardless of their modalities. Interestingly, toothbrush and toothpaste are mapped almost on top of each other in the text modality showing semantic and syntax similarity. However, in the RGB and depth modality they are close but not on top of each other since they do not look the same. Also, we can see that apple and lemon are mapped close to each other in all modalities which suggests that our proposed EMMA learns some notion of the concept of fruits. These qualitative results show that our propose GEOMETRIC ALIGNMENT and EMMA have an interpretable latent space.

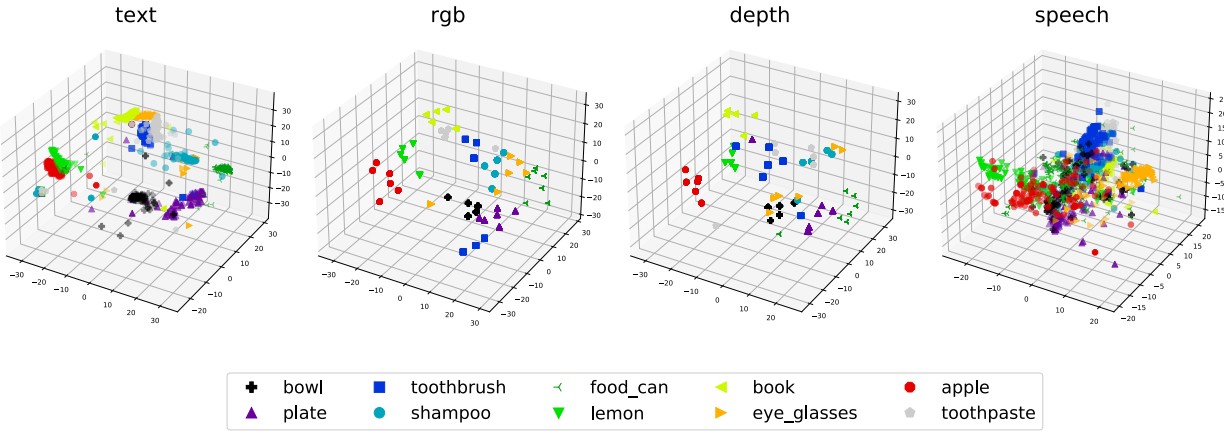

Figure 4: 3D T-SNE (Van der Maaten & Hinton, 2008) projection of test embeddings of 10 randomly selected classes of objects using EMMA. Each modality is separately projected into a three-dimensional space. Each RGB and depth image is associated with several language descriptions, leading to denser plots for text and speech. In a perfect embedding, all instances of a class would be clustered in identical areas of the embedding space across all modalities. We can see that EMMA successfully encourages all four modalities to live in a common manifold, allowing accurate retrieval even when modalities are missing.

An example of the need to consider multiple modalities jointly is shown in fig. 5, showing how EMMA is able to correctly select an object instance from several similarly shaped and describable objects.

## 6.1   Discussion

Our proposed model performs well and learns fast, has been demonstrated to handle four modalities of shared information effectively, and is robust to test-time situations where information from one or more modalities is missing. There remains room for improvement. Specifically, the speech modality is harder to

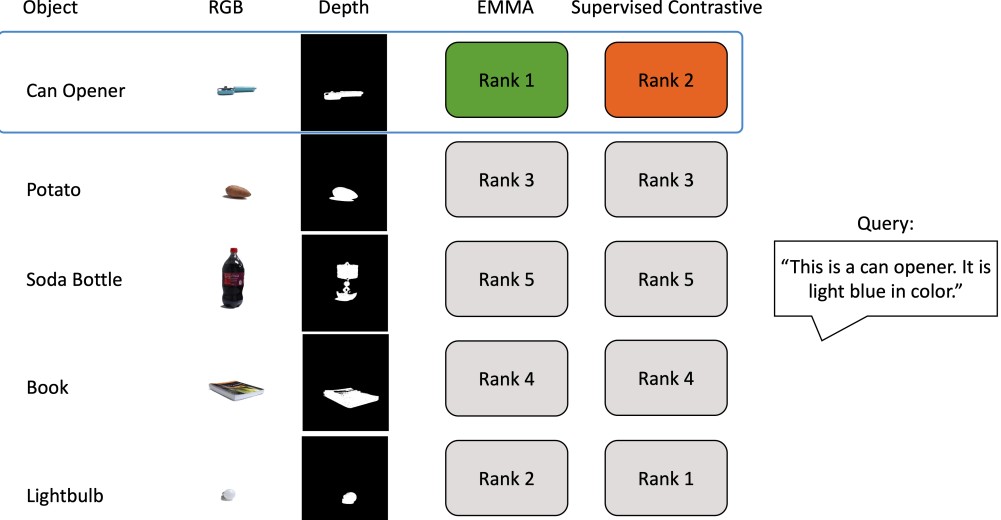

Figure 5: EMMA more accurately ranks objects and handles ambiguities with respect to a retrieval queries. Note that the phrase "light blue" in the query is very similar to "light bulb" and while SUPCON confuses this and predicts light bulb as the correct object, EMMA correctly identifies the can opener as the intended object, ranking the light bulb second.

handle. Figure 4 shows that although the relative position of instances are correct in the speech space, the distinction and clustering of different objects are not as good as the other three modalities.

Text seems to be the best clustered modality, and that makes sense because the variation in written text is much smaller than the other three modalities. Variation in speech is higher because there are a number of factors affecting speech understanding, including different accents, native language, gender, and age (Kebe et al., 2022). Variation in RGB and depth is higher than in text due to variations in lighting conditions, an object's texture and shape, the angle of the camera, and other factors.

## 7 Conclusion

In this work, we have demonstrated the effectiveness of a novel approach to learning from high-dimensional multimodal information even when one or more modalities is unavailable at test time. Our approach performs well on an object retrieval task from a testbed that contains four separate modalities, consistent with information that might be available to a physical agent, and outperforms state of the art contrastive learning approaches. Our proposed method is general enough to be applied to a variety of multimodal retrieval problems, and is not limited to purely language-based image retrieval.

In future, this work will be extended to solve less clearly delineated problems; for example, differentiating among members of a class, as well as across classes. However, this work represents a significant step towards handling such retrieval problems, while not arbitrarily limiting the number of sensor and other modalities that can be incorporated.

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
