# OpenReview forum: "Multimodal Language Learning in the Face of Missing Modalities"
_TMLR — Rejected by TMLR_

### Review · Reviewer_Aw92 · 2022-09-29

**Summary Of Contributions:**

This paper introduces a new multimodal alignment method that combines cross-entropy loss and contrastive loss. This loss can be used to learn retrieval models that incorporate an arbitrary number of modalities.

In particular, authors define a geometric alignment loss by minimizing the distance between each pair of positive points from heterogeneous modalities and maximizing the distance between each positive and negative points from all modalities simultaneously. The proposed EMMA loss is built on top of the geometric alignment loss and an additional cross-entropy loss.

Authors evaluate their proposed method on a multimodal dataset and use the mean reciprocal rank as their evaluation metric. Compared with other baselines, the results are mixed.

**Broader Impact Concerns:**

I do not see any.

**Requested Changes:**

1. In Section 5.1, the descriptions of training/validation/test splits are missing. The size of test set is important when evaluating the model on the retrieval task.

2. The details of pre-trained models are missing, such as BERT-base or BERT-large.

4. In figure 2, the speech modality is encoded by wav2vec2 instead of BERT?

5. There are some typos, such as wav2veq2 => wav2vec2.

**Strengths And Weaknesses:**

Strengths:

1. This paper extends the contrastive loss to multiple modalities including RGB, depth, text and speech. This task is motivated by real-world applications, such as robot navigation and computer-human interactions.

2. From table (a) we can see better results when text modality is available. In figure 3, the proposed method shows a faster converging speed compared to the baselines. In figure 4, the 3D T-SNE visualization shows that how embeddings from different modalities can be separated.

Weaknesses:

1. Compared to the baselines, the proposed method shows advantages only when the text modality is available. This limits the generalization ability of this method. In particular, the improvements of proposed geometric loss and EMMA loss seem to be marginal in table (a)(e.g., 0.09% of geometric loss and 0.69% of EMMA). The same is true for table (b).

2. In EMMA loss, the text modality is explicitly incorporated to encourage the alignments with other modalities. As the text modality is encoded using a pre-trained BERT, such design gives a strong prior to learn alignments faster and better. It is a little bit unfair when comparing other baselines.

3. There are some missing baselines in table (a) and (b). As EMMA is a combination of the geometric loss and cross-entropy loss, another baseline should be the combination of supervised contrastive loss and cross-entropy loss.

4. In general, both geometric loss and supervised contrastive loss are considering some supervised information from labels. The comparison of both methods does not show a clear advantage of the proposed loss, especially considering only speech/depth and speech/RGB modalities.

---

> ### Author Response · Authors · 2022-10-19
> **Response to reviewer Aw92**
>
> We thank the reviewer for the comments and suggestions, and here we attempt to clarify the points raised. As per the TMLR guidelines, we will update the paper when all reviews are present; however, in order to minimize wait time, we respond to this review before then.
>
> Weaknesses 1 and 4. We agree with the reviewer that the final converged performance of EMMA and SupCon are very similar. We note, however, that EMMA converges significantly faster (almost 5 times faster), and that is the main advantage of our method. This can be seen in Fig. 3. We would also like to note that, while EMMA performs best when the text modality is present, the baseline SupCon performs best when speech is present, suggesting that each approach has distinct advantages in different environments.
>
> Weakness 3. We apologize for the confusion, and appreciate the chance to clarify. We view SupCon as a form of cross-entropy loss: that is, EMMA is the combination of the geometric loss and SupCon as mentioned in the section “Combining Geometric and Cross-Entropy Losses” right above equation 8. Therefore, we considered all experiments, including geometric loss, SupCon loss, and geometric loss + SupCon loss (which we refer to as EMMA). To resolve this confusion, we will modify our wording in the paper from *cross-entropy loss* to *cross-entropy based SupCon loss*.
>
> Weakness 2. Our approach, EMMA, treats all modalities in the same way and does not anchor on any specific modality. Referring to EMMA’s objective function (equation 8), this means that indices *j*, *k*, and *m* iterate over all available modalities for each instance i. We will clarify this in the paper by adding the following to the discussion of equation 8:
>
> > “Both the geometric alignment and cross-entropy portions of EMMA are designed to not anchor on any specific modality, but instead consider all available modalities. This geometric portion is in contrast to previous triplet-based approaches. To show this, we consider how the geometric portion of EMMA’s loss would apply to the example in Figure 2. Considering the apple (on the left) as instance *i*, the dotted lines connecting the embeddings of the apple’s modalities reflect minimizing the pairwise distances between the different modalities of *i*: in equation 8, we do this via the h(x, y) function. The dashed lines connecting the representations of the apple to a *not apple* (the coffee mug) reflect maximizing the distances between these embeddings (done via g(x,y). Notice how in both cases we consider all relevant pairs.”
>
> Additionally, we would like to note we are using pre-trained BERT for the text modality in SupCon as well as EMMA, so the comparison is not unfair in that sense. While EMMA does perform less well when the text modality is ablated, it still converges faster.
>
> **Response to Requested Changes**:
> 1. Our training, validation, and test splits were 7380, 4160, and 4960 respectively. We will add this information in section 5.1 of the paper.
> 2. For our pre-trained models, we used “Document Pool Embeddings” from flair which performs an average pooling over all embeddings computed by BERT-base-uncased. We will add this information in section 4.3.
> 3. Thank you for catching this. We will change BERT to wav2vec2 in figure 2 for the speech modality.
> 4. We will do a thorough read-through and correct typos found. When all reviews are present we will respond with the updated version.

---

### Review · Reviewer_qVnU · 2022-10-17

**Summary Of Contributions:**

This paper presents EMMA, a method for learning joint representations of multiple modalities (images, text, speech) that are robust to inputs with missing modalities. EMMA integrates contrastive learning with a geometric alignment loss to learn representations that minimize distances between instances and maximize distances between classes. Unlike other methods, EMMA can easily incorporate additional modalities, and is 5x faster to train than existing baselines. Experiments are conducted on the GoLD dataset, which consists of simple language-grounding classification tasks with multi-modal inputs: RGB images, depth images, text descriptions, and speech descriptions. Results measuring classification accuracy indicate that EMMA is marginally better or similar to prior methods like Contrastive learning and Supervised Contrastive learning.


**Broader Impact Concerns:**

The reviewer does not see any major ethical or moral concerns regarding the method or dataset used in the paper.


**Requested Changes:**

Based on the comments above:
- Clarify the benefits of EMMA’s ~1% gain over Contrastive and Supervised Contrastive learning in a realistic setting with agents.
- Explain why training speed matters with concrete problem scenarios.
- Define concepts before they are used, and make the writing less verbose.
- Maybe evaluate on a better dataset/environment that involves realistic (or simulated) embodied environments, rather than just evaluating on 1-of-N classification tasks.


**Strengths And Weaknesses:**

Strengths
- EMMA tackles an important problem in real-world perception. Robots and other embodied agents in physical environments often suffer from sensory failures and occlusions. Quite often, embodied agents rely too much on a single modality like vision, which can be risky. Learning joint representations of multiple modalities that are robust to modality dropouts could potentially improve the reliability of such systems.
- The experiments include a (mostly) good set of baselines and modality ablations. The reviewer is not too familiar with self-supervised learning losses, but contrastive and triplet losses seem quite prevalent in recent pre-training approaches like CLIP, MoCo, MERLOT etc. The MRR metric used for evaluations also seems reasonable for classification tasks.
- Figures 3 and 4 are particularly informative. The qualitative results in Figure 4 could be further explored in future works to study the generalization capabilities of multi-modal representations over uni-modal representations.

Weaknesses
- EMMA achieves marginally better or comparable accuracy to simpler and more established methods like Contrastive and Supervised Contrastive loss (see Table 1). In several places, the paper makes strong assertions that EMMA “outperforms state-of-the-art” approaches, but the gains in Table 1 are so small, especially after accounting for error ranges – the use of “outperforms” feels unsubstantiated. In fact, in 6/16 modality ablations in Table 1, Supervised Contrastive learning outperforms EMMA. And even for ablations for which EMMA achieves the highest accuracy, the gains are within a 0.5-1% margin of other baselines. Going back to the original use-case of agents dealing with missing modalities, what does 0.5-1% better classification accuracy correspond to? Would an agent with faulty sensors be substantially better off with a more complicated approach that achieves marginal gains?
- The key benefit of EMMA over other methods is that EMMA can be trained 5x faster, but was training speed the original motivating issue? In Figure 3, EMMA saturates 2^5 - 2^3 = 24 epochs before other methods. But how long does each epoch take? And is training speed the real bottleneck for agents with missing modalities? Since all methods are trained offline, it’s unclear what are the benefits of faster training speeds, or at least it’s hard to connect this feature back to the original motivation, which is about learning robust representations despite missing modalities.
- In general, the writing can be greatly improved. More specifically:
  - Concepts and jargons are often used before they are defined. In the first line of the paper “We propose EMMA, a generalized geometric method combined with a cross-entropy loss function …” – a first time reader might wonder what is a “geometric method” or what does it mean to “generalize it”, or even why “cross-entropy loss” is being mentioned in the opening line. Perhaps starting with “We propose EMMA, a framework for learning multi-modal representations that are robust to missing modalities. Our method works by …” would improve readability. Similarly, the Positive/Negative/Anchor definitions in page 7 are defined after they are used in the prose. Similar examples are sprinkled throughout the paper.
  - The writing is a bit verbose. TMLR doesn’t have page limits, but 15 pages might be a bit too much for a paper that combines existing loss functions, and evaluates on a single dataset. The introduction and related work paragraphs could be greatly shortened by focusing on the relevant problem: “learning multimodal representations that are robust to missing modalities”.
- Despite being motivated by agents with faulty sensors, the choice of the GoLD dataset – a static classification dataset – seems odd. For embodied agents, classifying among 5 possible objects is rarely a relevant problem formulation.

---

> ### Author Response · Authors · 2022-11-04
> **Response to Reviewer qVnU**
>
> We thank the reviewer for the comments and suggestions, and here we attempt to clarify them.
>
> W1: We agree with the reviewer that the results are, for the most part, comparable to our baseline once converged. However, the main contribution of the paper is to extend and modify existing methods to be able to incorporate any number of modalities. As originally proposed, the SupCon approach which we chose as our baseline was originally applied to unimodal datasets such as ImageNet, CIFAR-10, and CIFAR-100. We both demonstrate that it is possible to use on multimodal datasets and extend it with the addition of geometric approaches, which gives benefits on training speed and performance when modalities are dropped. As such, we believe that this work meets TMLR’s emphasis on technical correctness (as compared to subjective significance). We will be sure to add this information in the paper to better clarify our contribution.
>
> W2: We acknowledge that the bottleneck for agents in different settings may differ, and training speed may not be critical in offline learning scenarios. However, since we usually need to finetune models for different tasks when it comes to transfer learning, the training speed becomes relevant. We note that faster convergence is not the main goal of this work, but rather an outcome of our considered model design. The main contribution of this work is to propose a general multimodal learning approach that can incorporate any number of modalities.
>
> W3: We will do a thorough proofread and clarify places in the writing where it is appropriate to reorder definitions and use of terminology. We will also seek ways in which the overall length of the paper can be reduced without reducing its comprehensibility; however, our goal is to balance the broader research context with concise writing, particularly for people who may be newer to the research topic. Would the reviewer be able to provide an assessment on whether a reduction in length is worth not providing this additional context?
>
> W4: To replicate faulty sensors, we either completely drop a modality as reported in the paper, or apply random dropouts on a modality. Also, we ran experiments where we do not use a modality for training at all. We will add these results in order to clarify and strengthen the paper.

---

### Review · Reviewer_e9X5 · 2022-10-19

**Summary Of Contributions:**

This paper proposes extended multimodal alignment (EMMA) to learn multimodal retrieval models that incorporate an arbitrary number of views and under the setting where modalities can be unavailable. The authors propose an approach combining contrastive learning to align modalities with explicit geometric alignment, by arguing that the two are complementary. Experiments on grounded language object retrieval based on four modalities including vision, depth sensing, text, and speech, shows strong results and faster convergence.

**Broader Impact Concerns:**

none.

**Requested Changes:**

See weaknesses above, especially empirically or theoretically justifying the complementarity of both alignment objectives, more analysis into learning with both objectives (perhaps with reference to multitask learning literature), and better results are needed.

**Strengths And Weaknesses:**

Strengths:
1. The paper is well motivated and studies a relevant problem.
2. Paper is well written and mostly clear.

Weaknesses:
1. The paper's main claim hinges on the complementarity of both alignment objectives, where the authors argue that 'In Geometric Alignment, the advantages are an intuitive learning objective in terms of distance, interpretability of the learned embedding space, and faster convergence. The advantage of SupCon is that it uses a classification objective which is aligned with the downstream task.', so they proposed to combine both. This argument is rather hand-wavey - for example, I do not see any reason for why the former would lead to a more interpretable representation space while the latter would not. Also, the results do not fully justify the fact that the former converges faster in all cases. These 2 objective functions seem largely similar to me especially when cosine similarity is used as the distance function, the only difference being whether margin loss or contrastive loss is used. A much better justification of the differences between the optimization process of these 2 objective functions is needed to justify their combination as a main contribution of the paper.
2. In the results in Table 1 it is not clear that the combination of both objectives significantly outperforms either alone, since the standard deviations overlap.
3. There should be a deeper analysis of the computational tradeoffs since you are computing and optimizing 2 objective functions, and how does that affect learning.
4. There can also be deeper analysis of how learning should be performed - ablating training for both at the same time, or 1 then the other?

---

> ### Author Response · Authors · 2022-11-04
> **Response to reviewer e9X5**
>
> We thank the reviewer for the comments and suggestions, and here we attempt to clarify them.
>
> W1: While both SupCon and the geometric approach are trying to bring target objects together (and push dissimilar ones away), there is a tradeoff in SupCon where you have a normalized ranking across them. In contrast, in geometric approaches distances can be arbitrarily near or far. Our approach tries to respect distinct modalities of each instance. However, the main contribution of the paper is to extend and modify existing methods to be able to incorporate any number of modalities. As originally proposed, the SupCon approach which we chose as our baseline was originally applied to unimodal datasets, and we both demonstrate that it is possible to use it on multimodal datasets, and extend it with the addition of geometric approaches, which gives benefits on training speed and performance when modalities are dropped.
>
> We note that while our formulation is similar to SupCon notationally, conceptually SupCon uses different augmentations of RGB images as positive examples (inherently coming from the same underlying empirical distribution), while we use different modalities drawn from different distributions. To the best of our knowledge our work is the first to use supervised contrastive learning for such a multimodal scenario, where the different language and sensor inputs provide more robust learning than can be accomplished with single sensor inputs, even augmented. As such, we believe that this work meets TMLR’s emphasis on technical correctness (as compared to subjective significance).
>
> W2: We agree that the combination of the two objectives results, after training, in comparable performance. However, as concerns the contributions of the paper, we focus on (1) the improved speed of convergence during training, and (2) the appropriateness of our method for multimodal learning problems: we propose a general loss function that can take any number of modalities and align them. Our proposed model can be applied to multimodal datasets with minimal modifications while achieving state of the art performance. We note that SupCon itself, as proposed, has been used for unimodal datasets such as ImageNet, CIFAR-10, and CIFAR-100, while the dataset we use includes four modalities.
>
> W3: In section 4.1.2, we provide some analysis of the time complexity of the two approaches based on the number of available modalities and the batch size. We will clarify this, and extend this discussion with timing data, as follows:
>
> EMMA takes almost 8 epochs to converge and each epoch takes roughly 0.7 minutes which makes it 5.6 minutes until convergence, while SupCon takes about 36 epochs to converge and each epoch takes 0.52 minutes which amounts to 18.72 minutes. That is when we use all four modalities for training. When we ablate one or two modalities, the training takes less time.
>
> W4: In this note, is the reviewer referring to ablating objectives or modalities? If the reviewer is referring to the former, the three cases we considered cover all the possible scenarios; ablating SupCon objective, ablating geometric objective or ablating none of them. If the latter is intended, we have performed experiments on ablating modalities during training, which is a rather large set of combinatorial results. Broadly speaking, the takeaway from ablating modalities during training time is similar to ablation during testing, in that ablating text worsens the results; and ablating both text and speech results in a null query. Also, testing with a modality that was ablated during training results in a random guess. E.g. if we ablate depth during training, the model cannot perform well on the test data when we use text, speech, and depth, though it still performs reasonably well if we use all four modalities and it learns to take advantage of the RGB modality. If the reviewer could clarify whether this is the desired topic of discussion we will clarify this in the paper.

---

### Decision · Action_Editors · 2022-12-04

**Recommendation:** Reject

**Comment:**

I encourage the resubmission of an updated manuscript that tackles several of the open questions by the reviewers.  In particular, while it is obviously simplest when one method works best overall, a rich analysis that provides insight into when these modalities are complementary (or not) and whether certain losses are best in those cases would also be appreciated.

**Audience:**

The audience is anyone interested in multimodality.   I am very sympathetic to the aims of the paper, and the potential robustness of representations that accurately aggregate meaning from multiple sensors/views.  This should be broadly of interest to the machine learning community, but particularly those at the intersections of language, vision, and/or robotics.

**Claims And Evidence:**

The authors introduce a novel mechanism for learning multimodal representations -- specifically, "geometric alignment" which generalizes triplet style losses to multiple modalities.  The authors also present a very nice condition for learning in which an object/concept may be represented in RGB, Depth, Text, or Audio.  Results are then compared for subsets of modalities using their approach versus other losses.

The argument in the paper relies on both performance and training efficiency (performance / epoch).  The approach shows more promise on the latter metric than the former.

Fundamentally, there is no clear trend in terms of which loss function is best in any given subset of modalities, and often the "best" performance is within the std dev of another. This left reviewers unsure of what they learned from the results -- it's possible that something about interactions between specific modalities could be divined would which help with some of e9X5's concerns?  Insight might be gleaned from where EMMA learns faster than others?

In general, all reviewers felt the results were incomplete